# Highly Selective Cyclization and Isomerization of Propargylamines to Access Functionalized Quinolines and 1-Azadienes

**DOI:** 10.3390/molecules28176259

**Published:** 2023-08-26

**Authors:** Zheng-Guang Wu, Hui Zhang, Chenhui Cao, Chaowu Lu, Aiwei Jiang, Jie He, Qin Zhao, Yanfeng Tang

**Affiliations:** 1School of Chemistry and Chemical Engineering, Nantong University, Nantong 226019, China; 2Anhui Sholon New Material Technology Co., Ltd., Chuzhou 239500, China

**Keywords:** high selectivity, cyclization, isomerization, heterocycle, propargylamine

## Abstract

Developing new organic reactions with excellent atom economy and high selectivity is significant and urgent. Herein, by ingeniously regulating the reaction conditions, highly selective transformations of propargylamines have been successfully implemented. The palladium-catalyzed cyclization of propargylamines generates a series of functionalized quinoline heterocycles, while the base-promoted isomerization of propargylamines affords diverse 1-azadienes. Both reactions have good functional group tolerance, mild conditions, excellent atom economy and high yields of up to 93%. More importantly, these quinoline heterocycles and 1-azadienes could be flexibly transformed into valuable compounds, illustrating the validity and practicability of the propargylamine-based highly selective reactions.

## 1. Introduction

Modern organic reactions utilizing simple precursors that result in the generation of functional and structural diversity are highly attractive for the synthesis of specific molecules, such as natural products, pharmaceuticals and materials. Moreover, the high selectivity of organic reactions is still a subject of heightened concern, especially for multiple possible reactive sites on one substance. Therefore, how to exquisitely control the reaction activity to achieve a highly selective transformation is a significant challenge.

Propargylamines, a versatile class of compounds with unique chemical structures, consist of amine groups and alkyne moieties on the same backbone (Figure 1) [1]. Compounds with a carbon−carbon triple bond have special reactivity, which can behave both as electrophilic reagents and as a source of electrons in nucleophilic reactions [2]. In addition, the amine moiety of the propargylamine can undergo nucleophilic reactions. This unique characteristic allows propargylamine compounds to act as both electrophilic and nucleophilic substrates in a variety of chemical transformations, such as metal-catalyzed coupling, addition, cycloaddition etc. [3,4,5,6,7,8,9,10,11,12]. It is well known that propargylamines feature wonderful reaction activities and are used as building blocks in manufacturing different organic substrates, natural products and drug candidates, showing broad applications in many fields of chemistry [13,14,15,16,17,18]. Therefore, the development of novel propargylamine-based synthetic methodologies and the construction of functionalized heterocycles or valuable synthetic intermediates are highly desirable, despite significant progress in the functionalization of propargylamines using Au, Ag, Cu, Fe, Hg, microwave, superbase, etc. As shown in Figure 1 [19,20,21,22,23,24,25,26,27,28,29,30], the development of novel, efficient and practical approaches using mild reaction conditions from easily accessible precursors to enable the highly selective transformation of propargylamines by ingeniously controlling the reaction activity is still a challenging task.

In this study, propargyalmine modules from the Cu-catalyzed A3-coupling of very simple and easily available ingredients, amines, aldehydes and alkynes, are employed for investigating new transformations [31,32,33]. By ingeniously modulating the reaction conditions, the highly selective cyclization and isomerization of propargylamine have been successfully implemented (Figure 1). Using palladium catalyst, diverse quinolines are obtained, which are widely applicable in drug discovery and material science [34,35,36,37,38,39,40]. Alternatively, in the presence of a base, a series of synthetically valuable 1-azadienes has been smoothly prepared (Figure 1) [41,42,43,44].

## 2. Results and Discussion

The Cu(I)-catalyzed A3-coupling of a rich variety of precursors, such as amines, aldehydes and alkynes, was developed by Li and co-workers in 2002 and represents a general and efficient strategy for the synthesis of propargylamines [31]. Herein, to assess the reactivity, propargylamine **1a**, which is easily obtained from the A3-coupling of aniline, benzaldehyde and p-tolylacetylene, was selected as a model for selective transformation under different conditions.

At the outset, several commonly used metal catalysts, including palladium, copper, iron and nickel salts, were examined for this reaction, and Pd(OAc)_2_ exhibited the best reaction activity to solely generate **2a** with 65% yield (Table 1, entries 1–9). Subsequently, by screening different solvents (DMSO, NMP, DCE, dioxane, CH_3_CN and toluene), toluene found to be the most suitable choice, with 80% yield for **2a** (Table 1, entries 10–15). Remarkably, the reaction yield of quinoline **2a** was greatly reduced by adding TBAI or bases, and a new compound, 1-azadiene **3a**, was separated from the reaction system and characterized by NMR and HRMS analyses (Table 1, entries 16–19). The unexpected behavior of the reaction prompted us to deeply investigate the reaction conditions for 1-azadiene formation from propargylamine isomerization. In the presence of Cs_2_CO_3_, a series of solvents was evaluated. Notably, CH_3_CN exhibited a compelling advantage in this isomerization, with a yield of 81% (Table 1, entries 20–24). Further screening of various bases illustrated that Bu_4_NOAc is the most suitable additive for propargylamine isomerization, with a yield of up to 91% (Table 1, entries 25–29). By systematically modulating the reaction conditions, the rule for the highly selective cyclization and isomerization of propargylamine was successfully mastered. From the palladium-catalyzed cyclization, quinolines were smoothly obtained, whereas in the presence of Bu_4_NOAc, 1-azadienes could be ingeniously prepared via an isomerization process.

With the optimum reaction conditions established, the generality of the method was investigated in detail. Initially, the scope of the palladium-catalyzed cyclization of propargylamine was explored with respect to the different units of amine, aldehyde and alkyne in the propargylamine structure. As shown in Figure 2a, a wide range of substituents on the aromatic rings displayed good tolerance under the optimized condition. Propargylamines with either electron-donating (methyl, *tert*-butyl, methoxyl and phenyl) or electron-withdrawing (fluorine, chlorine and trifluoromethyl) groups can smoothly generate the corresponding quinolines in moderate to good yields. Notably, the aliphatic group (cyclohexyl, **1o**) and heteroarene (2-thiophenyl, **1s**)-substituted propargylamines are also efficient for this Pd-catalyzed cyclization to afford the expected products in 81% and 72% yields, respectively. More importantly, the gram-scale reaction of propargylamine **1m** (4 mmol) efficiently proceeded to afford the quinoline compound **2m** in a decent yield of 78% (Figure 2).

As for Bu_4_NOAc-promoted isomerization, the scope of substrates was also evaluated using various propargylamines with different substitution patterns and electronic properties. As shown in Figure 2b, both electron-donating (methyl, *tert*-butyl, phenyl, naphthyl and methoxyl) and electron-withdrawing (fluorine, chlorine and bromine) groups on the aromatic rings of propargylamines were well tolerated in this isomerization system, affording the corresponding 1-azadienes in excellent yields (81%~93%). It is noteworthy to mention that the halide and methoxyl groups on these products can be further functionalization in preparing other complex and diverse molecules.

To further explore the synthetic utility of this protocol, we employed one of the quinoline **2m** as the novel cyclometalated main ligand for constructing the highly efficient red phosphorescent iridium(III) complex **Ir-2m**, which shows bright red emission with a peak at 605 nm, high photoluminescence quantum yield (PLQY) of 70.12% and a small full width at half maximum (FWHM) value of 46 nm in CH_2_Cl_2_, illustrating the potential application in pure red OLED (Figure 1a,b and Appendix A) [45,46]. The application of various isomerization products was also explored. Several functionalized 1-azadienes synthesized in situ from propargylamines have been used to prepare a series of medicinally important dihydropyridin-2(*1H*)-ones (**4a**–**4f**) via a [4+2]-formal cycloaddition reaction with homophthalic anhydride under very simple reaction conditions with excellent functional tolerance and good yields (Figure 1c) [47,48].

Based on the above experimental results and previous studies on the functionalization of propargylamines [19,20,21,22,23,24,25,26,27,28,29,30], we also proposed a plausible mechanism for the Pd or base promoting highly selective cyclization and isomerization of propargylamines, as depicted in Figure 3. Initially, Pd(OAc)_2_ coordinates with the triple bond (**A1**) to enhance the electrophilicity of the alkyne part of the propargylamine. The subsequent intramolecular nucleophilic attack by the *N*-substituted aromatic ring generated intermediate **A2**. Protonolysis of the resulting intermediate **A2** gives dihydroquinoline **A3** and releases the palladium catalyst for a new cycle. Then, the generated dihydroquinoline was oxidized by O_2_ to afford the corresponding quinoline product **2**. In the presence of Bu_4_NOAc, propargylamine **1** first forms allenic intermediate **B1**, which subsequently undergoes a prototropic isomerization to obtain 1-azadiene **3**.

## 3. Materials and Methods

### 3.1. General Information

Commercially available reagents were used as received without purification. Raw materials were purchased from Bidepharm and Energy-chemical. Column chromatography was carried out using silica gel (200–300 mesh). Analytical thin–layer chromatography was performed on glass plates of Silica Gel GF–254 using UV detection. ^1^H, ^13^C and ^19^F NMR spectra were recorded on a Bruker AVANCE 400M spectrometer (Bruker, Billerica, MA, USA). The chemical shift references were as follows: ^1^H NMR (CDCl_3_) 7.26 ppm; ^13^C NMR (CDCl_3_) 77.0 ppm. HRMS spectra were obtained using Micromass GCT (ESI). The photoluminescence spectra were measured using a Hitachi F-4600 photoluminescence spectrophotometer (Hitachi, Kyoto, Japan). The absolute photoluminescence quantum yields (Φ) were measured using a HORIBA FL-3 fluorescence spectrometer (HORIBA, Kyoto, Japan).

### 3.2. Experimental Section

#### 3.2.1. Synthesis of Propargylamines following Reported Procedures (J. Org. Chem. 2006, 71, 2064–2070; Org. Lett. 2006, 8, 2405–2408; Tetrahedron 2014, 70, 3134–3140)

In a solution of amine (1.0 mmol), aldehyde (1.0 mmo) and alkyne (1.0 mmol) in DCM (10.0 mL), the reaction mixture was stirred at room temperature under nitrogen for 12 h. After removing the solvent using vacuum distillation, the crude mixture was purified via flash column chromatography to obtain the target product propargylamine.

#### 3.2.2. General Procedure for the Preparation of Quinolines through Palladium-Catalyzed Cyclization

In a solution of propargylamine (0.1 mmol) and Pd(OAc)_2_ (5 mol%) in toluene (2 mL), the reaction mixture was stirred at 80 °C under air for 12 h. After removing the solvent using vacuum distillation, the crude mixture was purified via flash column chromatography to obtain the target product.

#### 3.2.3. General Procedure for the Preparation of 1-Azadienes via Bu_4_NOAc-Promoted Isomerization

In a solution of propargylamine (0.1 mmol) and Bu_4_NOAc (0.2 mmol) in CH_3_CN (2 mL), the reaction mixture was stirred at 80 °C under air for 12 h. After removing the solvent using vacuum distillation, the crude mixture was purified via flash column chromatography to obtain the target product.

#### 3.2.4. General Procedure for the Preparation of **Ir-2m**

The mixture of **2m** (1.0 mmol) and IrCl_3_ (0.4 mmol) in 2-ethoxyethanol and water (20 mL, 3:1, *v*/*v*) was stirred at 130 °C for 24 h under argon. After cooling, the solid precipitate was filtered to obtain a crude cyclometalated Ir(III) chloro-bridged dimer. Then, the slurry of crude chloro-bridged dimer, Na_2_CO_3_ (5.0 mmol) and TMHD (5.0 mmol) in 2-ethoxyethanol (30 mL) was reacted at 120 °C for 24 h. The solvent was evaporated at low pressure, and the mixture was poured into water. Next, the mixture was extracted using CH_2_Cl_2_ and chromatographed to obtain the complex **Ir-2m** with a 66% yield.

#### 3.2.5. General Procedure for the Preparation of Dihydropyridin-2(1H)-Ones via Cycloaddition Reaction with 1-Azadienes and Homophthalic Anhydride

In a solution of propargylamine (1.0 mmol) and Bu_4_NOAc (2.0 mmol) in CH_3_CN (10 mL), the reaction mixture was stirred at 80 °C under air for 12 h. After cooling to room temperature, homophthalic anhydride (1.0 mmol) was added, and the mixture was stirred at room temperature under air for 12 h. After removing the solvent using vacuum distillation, the crude mixture was purified via flash column chromatography to obtain the target product.

### 3.3. Characterization of Products

phenyl-4-(*p*-tolyl)quinoline (**2a**): White solid, 23.7 mg, 81% yield (Eluent: petroleum ether/ethyl acetate = 50/1). ^1^H NMR (400 MHz, CDCl_3_) δ 8.24 (d, *J* = 8.4 Hz, 1H), 8.22–8.15 (m, 2H), 7.97–7.84 (m, 1H), 7.81 (s, 1H), 7.73 (ddd, *J* = 8.4, 6.8, 1.5 Hz, 1H), 7.57–7.39 (m, 6H), 7.36 (d, *J* = 7.8 Hz, 2H), 2.48 (s, 3H). ^13^C NMR (101 MHz, CDCl_3_) δ 156.9, 149.3, 148.8, 139.7, 138.4, 135.5, 130.1, 129.5, 129.5, 129.3, 128.9, 128.2, 127.6, 126.3, 125.9, 125.7, 119.4, 21.3. HRMS (ESI) *m/z* calcd for C_22_H_17_N [M+H]: 296.1439, found: 296.1439.

2,4-diphenylquinoline (**2b**): Light yellow solid, 19.7 mg, 72% yield (Eluent: petroleum ether/ethyl acetate = 50/1). ^1^H NMR (400 MHz, CDCl_3_) δ 8.25 (dd, *J* = 8.6, 1.3 Hz, 1H), 8.22–8.16 (m, 2H), 7.91 (dd, *J* = 8.5, 1.4 Hz, 1H), 7.82 (s, 1H), 7.74 (ddd, *J* = 8.4, 6.8, 1.5 Hz, 1H), 7.61–7.51 (m, 7H), 7.52–7.46 (m, 1H), 7.49–7.42 (m, 1H). ^13^C NMR (101 MHz, CDCl_3_) δ 156.9, 149.2, 148.8, 139.7, 138.4, 130.1, 129.6, 129.5, 129.3, 128.8, 128.6, 128.4, 127.6, 126.3, 125.8, 125.6, 123.5, 119.4, 115.9. HRMS (ESI) *m/z* calcd for C_21_H_15_N [M+H]: 282.1283, found: 282.1283.

methyl-2,4-diphenylquinoline (**2c**): Faint yellow solid, 23.4 mg, 78% yield (Eluent: petroleum ether/ethyl acetate = 50/1). ^1^H NMR (400 MHz, CDCl_3_) δ 8.21–8.10 (m, 3H), 7.78 (s, 1H), 7.65 (s, 1H), 7.56 (d, *J* = 4.4 Hz, 5H), 7.52 (dd, *J* = 8.2, 6.3 Hz, 3H), 7.49–7.40 (m, 1H), 2.48 (s, 3H). ^13^C NMR (101 MHz, CDCl_3_) δ 156.0, 148.5, 147.4, 139.8, 138.6, 136.3, 131.8, 129.8, 129.6, 129.2, 128.8, 128.6, 128.3, 127.5, 125.7, 124.4, 119.4, 21.8. HRMS (ESI) *m/z* calcd for C_22_H_17_N [M+H]: 296.1439, found: 296.1439.

6-(tert-butyl)-2,4-diphenylquinoline (**2d**): Light yellow solid, 23.6 mg, 70% yield (Eluent: petroleum ether/ethyl acetate = 40/1). ^1^H NMR (400 MHz, CDCl_3_) δ 8.23–8.13 (m, 3H), 7.90–7.76 (m, 3H), 7.63–7.39 (m, 8H), 1.35 (s, 9H). ^13^C NMR (101 MHz, CDCl_3_) δ 156.3, 149.2, 149.0, 138.6, 129.6, 129.6, 129.2, 128.8, 128.6, 128.4, 128.4, 127.5, 125.3, 120.5, 119.5, 35.1, 31.2. HRMS (ESI) *m/z* calcd for C_25_H_23_N [M+H]: 338.1909, found: 338.1910.

6-methoxy-2,4-diphenylquinoline (**2e**): Light yellow solid, 24.8 mg, 79% yield (Eluent: petroleum ether/ethyl acetate = 40/1). ^1^H NMR (400 MHz, CDCl_3_) δ 8.19–8.11 (m, 3H), 7.77 (s, 1H), 7.58 (s, 1H), 7.59–7.52 (m, 2H), 7.55–7.43 (m, 4H), 7.47–7.33 (m, 2H), 7.19 (d, *J* = 2.8 Hz, 1H), 3.80 (s, 3H). ^13^C NMR (101 MHz, CDCl_3_) δ 157.8, 154.7, 147.8, 144.9, 139.8, 138.7, 131.6, 129.4, 129.0, 128.8, 128.7, 128.4, 127.3, 126.7, 121.8, 119.7, 103.7, 55.5. HRMS (ESI) *m/z* calcd for C_22_H_17_NO [M+H]: 312.1388, found: 312.1388.

6-chloro-2,4-diphenylquinoline (**2f**): White solid, 22.3 mg, 73% yield (Eluent: petroleum ether/ethyl acetate = 50/1). ^1^H NMR (400 MHz, CDCl_3_) δ 8.22–8.14 (m, 3H), 7.89–7.82 (m, 2H), 7.67 (dd, *J* = 9.0, 2.3 Hz, 1H), 7.62–7.48 (m, 7H), 7.52–7.40 (m, 1H). ^13^C NMR (101 MHz, CDCl_3_) δ 157.1, 148.5, 147.2, 139.2, 137.8, 132.2, 131.7, 130.5, 129.6, 129.4, 128.9, 128.8, 128.7, 127.5, 126.5, 124.5, 120.1. HRMS (ESI) *m/z* calcd for C_21_H_14_ClN [M+H]: 316.0893, found: 316.0893.

2,4-diphenylbenzo[g]quinoline (**2g**): Light yellow solid, 23.2 mg, 71% yield (Eluent: petroleum ether/ethyl acetate = 40/1). ^1^H NMR (400 MHz, CDCl_3_) δ 8.27–8.19 (m, 2H), 8.13 (d, *J* = 9.0 Hz, 1H), 8.00 (d, *J* = 9.0 Hz, 1H), 7.88 (dd, *J* = 7.9, 1.5 Hz, 1H), 7.81 (s, 1H), 7.67 (d, *J* = 8.6 Hz, 1H), 7.60–7.42 (m, 9H), 7.16 (ddd, *J* = 8.6, 7.0, 1.5 Hz, 1H). ^13^C NMR (101 MHz, CDCl_3_) δ 155.5, 149.8, 149.2, 143.0, 139.1, 132.9, 131.5, 129.8, 129.3, 129.3, 129.2, 128.9, 128.6, 128.4, 128.1, 128.1, 127.4, 126.5, 125.5, 122.8, 121.8. HRMS (ESI) *m/z* calcd for C_25_H_17_N [M+H]: 332.1439, found: 332.1439.

4-phenyl-2-(*p*-tolyl)quinoline (**2h**): Faint yellow solid, 20.9 mg, 73% yield (Eluent: petroleum ether/ethyl acetate = 50/1). ^1^H NMR (400 MHz, CDCl_3_) δ 8.23 (dd, *J* = 8.6, 1.2 Hz, 1H), 8.13–8.06 (m, 2H), 7.89 (dd, *J* = 8.4, 1.4 Hz, 1H), 7.82–7.68 (m, 2H), 7.60–7.40 (m, 6H), 7.33 (d, *J* = 8.1 Hz, 2H), 2.43 (s, 3H). ^13^C NMR (101 MHz, CDCl_3_) δ 156.9, 149.1, 148.8, 139.5, 138.5, 136.9, 130.1, 129.6, 129.5, 128.6, 128.4, 128.1, 127.5, 126.2, 125.7, 125.6, 119.2, 21.4. HRMS (ESI) *m/z* calcd for C_22_H_17_N [M+H]: 296.1439, found: 296.1438.

2-([1,1′-biphenyl]-4-yl)-4-phenylquinoline (**2i**): Light yellow solid, 28.7 mg, 81% yield (Eluent: petroleum ether/ethyl acetate = 40/1). ^1^H NMR (400 MHz, CDCl_3_) δ 8.32–8.23 (m, 3H), 7.99–7.85 (m, 2H), 7.83–7.65 (m, 5H), 7.58 (s, 2H), 7.57–7.51 (m, 2H), 7.53–7.44 (m, 3H), 7.50–7.31 (m, 2H). ^13^C NMR (101 MHz, CDCl_3_) δ 156.5, 149.2, 148.9, 142.1, 140.6, 138.5, 138.4, 130.1, 129.6, 128.8, 128.6, 128.5, 128.0, 127.6, 127.6, 127.2, 126.4, 125.8, 125.7, 119.3. HRMS (ESI) *m/z* calcd for C_27_H_19_N [M+H]: 358.1596, found: 358.1596.

(naphthalen-2-yl)-4-phenylquinoline (**2j**): Light yellow solid, 23.6 mg, 74% yield (Eluent: petroleum ether/ethyl acetate = 40/1). ^1^H NMR (400 MHz, CDCl_3_) δ 8.65 (d, *J* = 1.7 Hz, 1H), 8.41 (dd, *J* = 8.6, 1.8 Hz, 1H), 8.30 (dd, *J* = 8.6, 1.2 Hz, 1H), 8.04–7.84 (m, 5H), 7.76 (ddd, *J* = 8.4, 6.8, 1.5 Hz, 1H), 7.67–7.45 (m, 8H). ^13^C NMR (101 MHz, CDCl_3_) δ 156.7, 149.3, 148.9, 138.5, 136.9, 133.9, 133.5, 130.1, 129.6, 129.5, 128.8, 128.6, 128.6, 128.5, 127.7, 127.2, 126.7, 126.4, 126.3, 125.8, 125.7, 125.1, 124.5, 119.5. HRMS (ESI) *m/z* calcd for C_25_H_17_N [M+H]: 332.1439, found: 332.1439.

(4-fluorophenyl)-4-phenylquinoline (**2k**): Faint yellow solid, 23.7 mg, 78% yield (Eluent: petroleum ether/ethyl acetate = 40/1). ^1^H NMR (400 MHz, CDCl_3_) δ 8.25–8.15 (m, 3H), 7.91 (dd, *J* = 8.4, 1.4 Hz, 1H), 7.80–7.70 (m, 2H), 7.56 (s, 3H), 7.60–7.50 (m, 2H), 7.48 (ddd, *J* = 8.3, 6.8, 1.3 Hz, 1H), 7.26–7.16 (m, 2H). ^13^C NMR (101 MHz, CDCl_3_) δ 155.8, 149.4, 148.8, 138.3, 135.8, 130.0, 129.7, 129.6, 129.5, 129.4, 128.7, 128.5, 126.4, 125.7, 119.0, 115.9, 115.7. ^19^F NMR (376 MHz, CDCl_3_) δ -112.45. HRMS (ESI) *m/z* calcd for C_21_H_14_FN [M+H]: 300.1189, found: 300.1190.

3-phenyl-2-(4-(trifluoromethyl)phenyl)quinoline (**2l**): Light yellow solid, 24.6 mg, 72% yield (Eluent: petroleum ether/ethyl acetate = 50/1). ^1^H NMR (400 MHz, CDCl_3_) δ 8.36–8.30 (m, 2H), 8.27 (d, *J* = 8.4 Hz, 1H), 7.94 (dd, *J* = 8.5, 1.4 Hz, 1H), 7.84 (s, 1H), 7.78 (dd, *J* = 8.5, 6.8 Hz, 3H), 7.60–7.45 (m, 6H). ^13^C NMR (101 MHz, CDCl_3_) δ 138.1, 130.2, 129.9, 129.6, 128.7, 128.6, 127.9, 127.0, 126.1, 125.8, 125.8, 119.2, 29.7. ^19^F NMR (376 MHz, CDCl_3_) δ -62.56. HRMS (ESI) *m/z* calcd for C_22_H_14_F_3_N [M+H]: 350.1157, found: 350.1156.

6-(tert-butyl)-4-phenyl-2-(*p*-tolyl)quinoline (**2m**): Light yellow solid, 28.3 mg, 82% yield (Eluent: petroleum ether/ethyl acetate = 50/1). ^1^H NMR (400 MHz, CDCl_3_) δ 8.16 (d, *J* = 8.8 Hz, 1H), 8.11–8.04 (m, 2H), 7.88–7.79 (m, 2H), 7.77 (s, 1H), 7.62–7.55 (m, 3H), 7.58–7.46 (m, 2H), 7.32 (d, *J* = 8.0 Hz, 2H), 2.43 (s, 3H), 1.35 (s, 9H). ^13^C NMR (101 MHz, CDCl_3_) δ 156.3, 149.0, 148.8, 147.4, 139.2, 138.7, 137.1, 129.6, 129.6, 128.6, 128.3, 127.4, 125.2, 120.5, 119.3, 35.1, 31.2, 21.4. HRMS (ESI) *m/z* calcd for C_26_H_25_N [M+H]: 352.2065, found: 352.2065.

6-(tert-butyl)-2-(4-fluorophenyl)-4-phenylquinoline (**2n**): Light yellow solid, 24.9 mg, 71% yield (Eluent: petroleum ether/ethyl acetate = 50/1). ^1^H NMR (400 MHz, CDCl_3_) δ 8.21–8.12 (m, 3H), 7.88–7.80 (m, 2H), 7.73 (s, 1H), 7.62–7.52 (m, 4H), 7.52 (ddd, *J* = 6.6, 5.3, 2.6 Hz, 1H), 7.25–7.14 (m, 2H), 1.35 (s, 9H). ^13^C NMR (101 MHz, CDCl_3_) δ 165.0, 162.5, 155.3, 149.3, 149.2, 147.3, 138.6, 136.0, 136.0, 129.5, 129.4, 129.3, 128.6, 128.6, 128.4, 125.2, 120.6, 119.1, 115.9, 115.6, 35.1, 31.2. ^19^F NMR (376 MHz, CDCl_3_) δ -112.78. HRMS (ESI) *m/z* calcd for C_25_H_22_FN [M+H]: 356.1815, found: 356.1815.

2-cyclohexyl-4-phenylquinoline (**2o**): White solid, 20.2 mg, 81% yield (Eluent: petroleum ether/ethyl acetate = 50/1). ^1^H NMR (400 MHz, CDCl_3_) δ 8.10–8.01 (m, 1H), 7.79 (dd, *J* = 8.4, 1.4 Hz, 1H), 7.50–7.38 (m, 6H), 7.19 (d, *J* = 4.8 Hz, 2H), 2.04–1.95 (m, 3H), 1.85–1.80 (m, 3H), 1.58 (dd, *J* = 12.4, 3.4 Hz, 2H), 1.42–1.37 (m, 2H), 1.18 (s, 1H). ^13^C NMR (101 MHz, CDCl_3_) δ 166.4, 148.7, 148.2, 138.5, 129.6, 129.3, 129.2, 128.5, 128.3, 125.7, 125.6, 125.6, 119.9, 47.7, 32.9, 29.7, 26.6, 26.1. HRMS (ESI) *m/z* calcd for C_21_H_21_N [M+H]: 288.1752, found: 288.1752.

4-(4-methoxyphenyl)-2-phenylquinoline (**2p**): Light yellow solid, 23.3 mg, 75% yield (Eluent: petroleum ether/ethyl acetate = 50/1). ^1^H NMR (400 MHz, CDCl_3_) δ 8.24 (d, *J* = 8.5 Hz, 1H), 8.22–8.15 (m, 2H), 7.96 (dd, *J* = 8.4, 1.4 Hz, 1H), 7.80 (s, 1H), 7.73 (ddd, *J* = 8.4, 6.8, 1.5 Hz, 1H), 7.58–7.46 (m, 6H), 7.13–7.05 (m, 2H), 3.92 (s, 3H). ^13^C NMR (101 MHz, CDCl_3_) δ 159.9, 156.9, 148.9, 139.7, 130.8, 130.7, 130.1, 129.5, 129.3, 128.8, 127.6, 126.2, 126.0, 125.7, 119.3, 114.1, 55.4. HRMS (ESI) *m/z* calcd for C_22_H_17_NO [M+H]: 312.1388, found: 312.1388.

4-(4-fluorophenyl)-2-phenylquinoline (**2q**): Light yellow solid, 20.9 mg, 71% yield (Eluent: petroleum ether/ethyl acetate = 40/1). ^1^H NMR (400 MHz, CDCl_3_) δ 8.26 (dd, *J* = 8.5, 1.2 Hz, 1H), 8.25–8.15 (m, 2H), 7.88 (dd, *J* = 8.3, 1.4 Hz, 1H), 7.81 (s, 1H), 7.75 (ddd, *J* = 8.4, 6.8, 1.4 Hz, 1H), 7.58–7.41 (m, 5H), 7.35 (dt, *J* = 7.6, 1.3 Hz, 1H), 7.29 (dt, *J* = 9.4, 2.1 Hz, 1H), 7.26–7.12 (m, 1H). ^13^C NMR (101 MHz, CDCl_3_) δ 164.0, 161.6, 156.9, 148.8, 147.8, 147.8, 140.6, 140.5, 139.5, 130.3, 130.2, 129.8, 129.5, 128.9, 127.6, 126.6, 125.4, 125.4, 125.4, 125.3, 119.3, 116.8, 116.6, 115.5, 115.3. ^19^F NMR (376 MHz, CDCl_3_) δ -112.48. HRMS (ESI) *m/z* calcd for C_21_H_14_FN [M+H]: 300.1189, found: 300.1189.

4-(4-chlorophenyl)-2-phenylquinoline (**2r**): Light yellow solid, 22.2 mg, 71% yield (Eluent: petroleum ether/ethyl acetate = 50/1). ^1^H NMR (400 MHz, CDCl_3_) δ 8.25 (d, *J* = 8.4 Hz, 1H), 8.22–8.15 (m, 2H), 7.85 (dd, *J* = 8.4, 1.4 Hz, 1H), 7.81–7.65 (m, 2H), 7.58–7.43 (m, 8H). ^13^C NMR (101 MHz, CDCl_3_) δ 156.9, 148.8, 147.9, 139.5, 136.8, 134.7, 130.9, 130.3, 129.7, 129.5, 128.9, 127.6, 126.6, 125.5, 125.3, 119.3. HRMS (ESI) *m/z* calcd for C_21_H_14_ClN [M+H]: 316.0893, found: 316.0893.

2-phenyl-4-(thiophen-2-yl)quinoline (**2s**): Light yellow solid, 20.3 mg, 72% yield (Eluent: petroleum ether/ethyl acetate = 40/1). ^1^H NMR (400 MHz, CDCl_3_) δ 8.26 (ddd, *J* = 14.2, 8.5, 1.4 Hz, 2H), 8.21–8.14 (m, 2H), 7.92 (s, 1H), 7.75 (ddd, *J* = 8.3, 6.8, 1.4 Hz, 1H), 7.59–7.40 (m, 6H), 7.25 (t, *J* = 4.3 Hz, 1H). ^13^C NMR (101 MHz, CDCl_3_) δ 156.9, 149.0, 141.5, 139.4, 139.2, 130.2, 129.7, 129.4, 128.9, 128.5, 127.8, 127.6, 127.2, 126.7, 125.4, 125.3, 119.8. HRMS (ESI) *m/z* calcd for C_19_H_13_NS [M+H]: 288.0847, found: 288.0847.

(2E)-N,1-diphenyl-3-(*p*-tolyl)prop-2-en-1-imine (**3a**): Light yellow solid, 26.8 mg, 91% yield (Eluent: petroleum ether/ethyl acetate = 40/1). ^1^H NMR (400 MHz, CDCl_3_) δ 7.78–7.69 (m, 2H), 7.47 (dd, *J* = 4.1, 1.8 Hz, 2H), 7.38–7.34 (m, 2H), 7.22–7.08 (m, 6H), 6.93–6.85 (m, 2H), 2.34 (d, *J* = 11.3 Hz, 3H). ^13^C NMR (101 MHz, CDCl_3_) δ 167.3 (C=N), 151.0, 141.7, 132.9, 129.8, 129.5, 129.5, 129.4, 128.8, 128.3, 127.5, 127.5, 123.8, 120.8, 21.4. HRMS (ESI) *m/z* calcd for C_22_H_19_N [M+H]: 298.1596, found: 298.1594.

(2E)-N,1,3-triphenylprop-2-en-1-imine(**3b**): Light yellow solid, 25.3 mg, 88% yield (Eluent: petroleum ether/ethyl acetate = 40/1). ^1^H NMR (400 MHz, CDCl_3_) δ 7.75 (t, *J* = 2.0 Hz, 1H), 7.49 (q, *J* = 1.4 Hz, 3H), 7.37 (s, 1H), 7.30 (d, *J* = 9.6 Hz, 7H), 7.13–7.11 (m, 1H), 6.97 (d, *J* = 1.3 Hz, 1H), 6.92 (s, 1H). ^13^C NMR (101 MHz, CDCl_3_) δ 167.2 (C=N), 150.9, 141.7, 139.4, 135.7, 131.6, 129.4, 129.4, 128.9, 128.8, 128.4, 127.5, 124.0, 121.9, 120.8. HRMS (ESI) *m/z* calcd for C_21_H_17_N [M+H]: 284.1439, found: 284.1441.

(2E)-N-(4-(tert-butyl)phenyl)-1,3-diphenylprop-2-en-1-imine (**3c**): Light yellow solid, 28.3 mg, 85% yield (Eluent: petroleum ether/ethyl acetate = 30/1). ^1^H NMR (400 MHz, CDCl_3_) δ 7.74 (t, *J* = 2.0 Hz, 1H), 7.48 (t, *J* = 1.7 Hz, 2H), 7.39–7.30 (m, 8H), 6.98–6.92 (m, 3H), 1.36 (s, 9H). ^13^C NMR (101 MHz, CDCl_3_) δ 166.9 (C=N), 148.1, 146.9, 141.3, 139.7, 135.9, 132.0, 129.8, 129.4, 128.8, 128.3, 127.5, 125.7, 122.3, 120.8, 34.4, 31.5. HRMS (ESI) *m/z* calcd for C_25_H_25_N [M+H]: 340.2065, found: 340.2063.

(2E)-N-(3-bromophenyl)-1,3-diphenylprop-2-en-1-imine (**3d**): Light yellow solid, 32.7 mg, 92% yield (Eluent: petroleum ether/ethyl acetate = 40/1). ^1^H NMR (400 MHz, CDCl_3_) δ 7.71 (d, *J* = 2.0 Hz, 2H), 7.50 (s, 3H), 7.34–7.29 (m, 2H), 7.14 (t, *J* = 2.1 Hz, 2H), 6.94 (s, 2H), 6.86 (d, *J* = 2.6 Hz, 3H). ^13^C NMR (101 MHz, CDCl_3_) δ 168.0 (C=N), 152.4, 142.7, 138.9, 135.4, 131.1, 130.3, 129.3, 128.9, 128.4, 127.6, 126.8, 124.1, 123.6, 122.7, 121.3, 119.3. HRMS (ESI) *m/z* calcd for C_21_H_16_BrN [M+H]: 362.0544, found: 362.0546.

(2E)-N-(2-bromo-4-fluorophenyl)-1,3-diphenylprop-2-en-1-imine (**3e**): Light yellow solid, 34.2 mg, 91% yield (Eluent: petroleum ether/ethyl acetate = 40/1). ^1^H NMR (400 MHz, CDCl_3_) δ 7.80 (t, *J* = 1.6 Hz, 2H), 7.50 (d, *J* = 7.0 Hz, 5H), 7.32 (d, *J* = 11.7 Hz, 1H), 7.24 (d, *J* = 11.8 Hz, 2H), 6.95 (s, 1H), 6.87–6.84 (m, 1H), 6.73 (s, 1H). ^13^C NMR (101 MHz, CDCl_3_) δ 169.3 (C=N), 157.6, 142.9, 138.5, 135.4, 130.7, 130.2, 129.8, 129.3, 128.9, 128.4, 128.2, 127.7, 121.5, 121.1, 119.9, 115.1, 114.9. ^19^F NMR (376 MHz, CDCl_3_) δ -118.89. HRMS (ESI) *m/z* calcd for C_21_H_15_BrFN [M+H]: 380.0450, found: 380.0453.

(2E)-1-([1,1′-biphenyl]-4-yl)-N,3-diphenylprop-2-en-1-imine (**3f**): Light yellow solid, 32.1mg, 87% yield (Eluent: petroleum ether/ethyl acetate = 40/1). ^1^H NMR (400 MHz, CDCl_3_) δ 7.90–7.80 (m, 1H), 7.77–7.57 (m, 4H), 7.59–7.48 (m, 1H), 7.50–7.38 (m, 3H), 7.42–7.32 (m, 3H), 7.35–7.28 (m, 3H), 7.21–7.12 (m, 1H), 7.06–6.88 (m, 3H). ^13^C NMR (101 MHz, CDCl_3_) δ 166.8 (C=N), 150.9, 144.8, 142.8, 141.6, 140.5, 139.9, 135.7, 129.9, 129.4, 128.9, 128.8, 128.2, 127.5, 127.2, 127.1, 124.0, 122.0, 120.9. HRMS (ESI) *m/z* calcd for C_27_H_21_N [M+H]: 360.1752, found: 360.1754.

(2E)-1-(naphthalen-2-yl)-N,3-diphenylprop-2-en-1-imine (**3g**): Light yellow solid, 28.2 mg, 84% yield (Eluent: petroleum ether/ethyl acetate = 30/1). ^1^H NMR (400 MHz, CDCl_3_) δ 8.24 (d, *J* = 1.7 Hz, 1H), 7.96–7.91 (m, 3H), 7.71–7.68 (m, 1H), 7.40–7.24 (m, 9H), 6.99 (d, *J* = 7.6 Hz, 3H). ^13^C NMR (101 MHz, CDCl_3_) δ 167.1 (C=N), 151.0, 144.8, 141.7, 136.8, 135.7, 134.1, 129.4, 128.9, 128.8, 128.8, 127.7, 127.5, 126.4, 124.0, 122.2, 120.8. HRMS (ESI) *m/z* calcd for C25H19N [M+H]: 334.1596, found: 334.1593.

(2E)-1-(4-fluorophenyl)-N,3-diphenylprop-2-en-1-imine (**3h**): Light yellow solid,27.4 mg, 93% yield (Eluent: petroleum ether/ethyl acetate = 40/1). ^1^H NMR (400 MHz, CDCl_3_) δ 7.76 (d, *J* = 3.4 Hz, 1H), 7.37–7.24 (m, 6H), 7.16 (d, *J* = 1.7 Hz, 3H), 6.94 (t, *J* = 1.1 Hz, 2H), 6.89 (s, 2H). ^13^C NMR (101 MHz, CDCl_3_) δ 166.0 (C=N), 150.8, 141.6, 135.5, 131.4, 131.3, 129.5, 128.9, 128.9, 127.5, 124.1, 121.9, 120.8, 115.5, 115.3. ^19^F NMR (376 MHz, CDCl_3_) δ -110.92. HRMS (ESI) *m/z* calcd for C_21_H_16_FN [M+H]: 302.1345, found: 302.1347.

(2E)-1-(4-chlorophenyl)-N,3-diphenylprop-2-en-1-imine (**3i**): Light yellow solid, 28.5 mg, 90% yield (Eluent: petroleum ether/ethyl acetate = 40/1). ^1^H NMR (400 MHz, CDCl_3_) δ 7.65–7.59 (m, 2H), 7.46–7.23 (m, 6H), 7.21–7.15 (m, 1H), 7.07 (dt, *J* = 6.0, 2.6 Hz, 1H), 6.91–6.79 (m, 4H). ^13^C NMR (101 MHz, CDCl_3_) δ 165.0 (C=N), 149.6, 140.6, 136.8, 135.0, 134.4, 129.7, 129.4, 128.6, 127.9, 127.8, 127.6, 126.5, 123.1, 119.7. HRMS (ESI) *m/z* calcd for C_21_H_16_ClN [M+H]: 318.1050, found: 318.1046.

(2E)-N,1-bis(4-fluorophenyl)-3-phenylprop-2-en-1-imine (**3j**): Light yellow solid, 25.5 mg, 81% yield (Eluent: petroleum ether/ethyl acetate = 30/1). ^1^H NMR (400 MHz, CDCl_3_) δ 8.11–8.02 (m, 1H), 7.76–7.73 (m, 1H), 7.42 (d, *J* = 3.0 Hz, 1H), 7.33 (s, 4H), 7.22–7.15 (m, 2H), 7.10–6.99 (m, 2H), 6.90 (t, *J* = 6.1 Hz, 2H). ^13^C NMR (101 MHz, CDCl_3_) δ 166.7 (C=N), 146.8, 141.9, 135.4, 134.8, 131.3, 130.7, 129.7, 128.9, 127.5, 122.2, 121.7, 115.8, 115.5, 115.3. ^19^F NMR (376 MHz, CDCl_3_) δ -110.63, -119.57. HRMS (ESI) *m/z* calcd for C_21_H_15_F_2_N [M+H]: 320.1251, found: 320.1253.

(2E)-3-(4-methoxyphenyl)-N,1-diphenylprop-2-en-1-imine (**3k**): Light yellow solid, 28.3 mg, 92% yield (Eluent: petroleum ether/ethyl acetate = 30/1). ^1^H NMR (400 MHz, CDCl_3_) δ 7.74–7.69 (m, 2H), 7.52–7.47 (m, 3H), 7.36 (t, *J* = 7.8 Hz, 2H), 7.28–7.24 (m, 2H), 7.16–7.11 (m, 2H), 6.87–6.80 (m, 3H), 3.79 (s, 3H). ^13^C NMR (101 MHz, CDCl_3_) δ 167.5 (C=N), 160.7, 151.0, 141.5, 139.5, 129.7, 129.4, 129.0, 128.8, 128.6, 128.3, 123.8, 120.8, 119.7, 114.2, 55.3. HRMS (ESI) *m/z* calcd for C_22_H_19_NO [M+H]: 314.1545, found: 313.2672.

(2E)-3-(2-fluorophenyl)-N,1-diphenylprop-2-en-1-imine (**3l**): Light yellow solid, 27.0 mg, 89% yield (Eluent: petroleum ether/ethyl acetate = 40/1). ^1^H NMR (400 MHz, CDCl_3_) δ 7.78–7.70 (m, 1H), 7.54–7.46 (m, 2H), 7.42–7.35 (m, 2H), 7.27 (dq, *J* = 13.9, 3.2 Hz, 2H), 7.19–7.04 (m, 3H), 6.98 (ddd, *J* = 16.0, 8.7, 1.2 Hz, 3H), 6.88 (s, 1H). ^13^C NMR (101 MHz, CDCl_3_) δ 166.7 (C=N), 150.7, 140.2, 137.9, 133.0, 130.3, 130.3, 129.3, 128.9, 128.4, 124.2, 123.1, 120.7, 116.3, 116.1, 113.9, 113.7. HRMS (ESI) *m/z* calcd for C_21_H_16_FN [M+H]: 302.1345, found: 302.1342.

(2E)-3-(4-fluorophenyl)-N,1-diphenylprop-2-en-1-imine (**3m**): Light yellow solid, 26.8 mg, 86% yield (Eluent: petroleum ether/ethyl acetate = 40/1). ^1^H NMR (400 MHz, CDCl_3_) δ 7.76–7.70 (m, 1H), 7.56–7.44 (m, 2H), 7.41–7.21 (m, 4H), 7.20–7.04 (m, 3H), 7.02–6.86 (m, 4H). ^13^C NMR (101 MHz, CDCl_3_) δ 166.7 (C=N), 161.8, 150.7, 140.2, 137.9, 132.9, 130.0, 129.3, 128.9, 128.9, 128.4, 123.1, 120.7, 116.1, 113.7. ^19^F NMR (376 MHz, CDCl_3_) δ -112.69. HRMS (ESI) *m/z* calcd for C_21_H_16_FN [M+H]: 302.1345, found: 302.1348.

(2E)-3-(2-chlorophenyl)-N,1-diphenylprop-2-en-1-imine (**3n**): Light yellow solid, 28.1 mg, 86% yield (Eluent: petroleum ether/ethyl acetate = 40/1). ^1^H NMR (400 MHz, CDCl_3_) δ 7.83–7.72 (m, 2H), 7.51–7.49 (m, 2H), 7.38–7.35 (m, 3H), 7.29–7.23 (m, 2H), 7.17–7.12 (m, 2H), 7.00–6.94 (m, 2H), 6.84 (d, *J* = 16.5 Hz, 1H). ^13^C NMR (101 MHz, CDCl_3_) δ 166.8 (C=N), 150.9, 139.0, 137.5, 134.3, 134.1, 130.2, 129.9, 129.4, 128.9, 128.6, 128.4, 127.3, 127.0, 124.4, 124.1, 120.7. HRMS (ESI) *m/z* calcd for C_21_H_16_ClN [M+H]: 318.1050, found: 318.1042.

(2E)-N,1-diphenyl-3-(thiophen-2-yl)prop-2-en-1-imine (**3o**): Light yellow solid, 24.6 mg, 85% yield (Eluent: petroleum ether/ethyl acetate = 40/1). ^1^H NMR (400 MHz, CDCl_3_) δ 7.76–7.66 (m, 1H), 7.52–7.43 (m, 2H), 7.41–7.23 (m, 3H), 7.18–6.83 (m, 6H), 6.74–6.62 (m, 1H). ^13^C NMR (101 MHz, CDCl_3_) δ 166.8 (C=N), 150.8, 141.0, 139.2, 137.2, 134.4, 132.8, 132.1, 129.3, 128.8, 128.4, 124.0, 121.0, 120.8. HRMS (ESI) *m/z* calcd for C_19_H_15_NS [M+H]: 290.1003, found: 290.1001.

**Ir-2m**: Red solid, 710.4 mg, 66% yield (Eluent: petroleum ether/dichloromethane = 10/1). ^1^H NMR (400 MHz, CDCl_3_) δ 8.19 (d, *J* = 9.2 Hz, 2H), 7.84 (s, 2H), 7.68–7.59 (m, 4H), 7.58–7.45 (m, 10H), 7.29 (dd, *J* = 9.2, 2.3 Hz, 2H), 6.68 (d, *J* = 7.9 Hz, 2H), 6.55 (s, 2H), 4.75 (s, 1H), 1.96 (s, 6H), 1.18 (s, 18H), 0.56 (s, 18H). ^13^C NMR (101 MHz, CDCl_3_) δ 192.5, 167.7, 150.9, 148.2, 146.8, 146.8, 143.9, 137.5, 137.0, 136.5, 128.6, 127.6, 127.6, 127.4, 125.9, 123.8, 123.7, 120.7, 119.2, 115.7, 39.5, 33.7, 30.0, 26.8, 20.6. HRMS (ESI) *m/z* calcd for C_63_H_67_IrN_2_O_2_ [M+H]: 1077.4910, found: 1077.4910.

**4a**: Light yellow solid, 358.1 mg, 78% yield (Eluent: dichloromethane/methyl alcohol = 100/1). ^1^H NMR (400 MHz, DMSO) δ 13.07 (s, 1H), 7.84 (d, *J* = 7.7 Hz, 1H), 7.43 (t, *J* = 7.5 Hz, 1H), 7.38 (s, 1H), 7.34–7.16 (m, 10H), 7.15 (d, *J* = 4.2 Hz, 2H), 7.15–7.07 (m, 1H), 7.06 (d, *J* = 7.6 Hz, 2H), 5.58 (d, *J* = 4.3 Hz, 1H), 5.11 (d, *J* = 8.9 Hz, 1H), 4.35 (dd, *J* = 8.7, 4.2 Hz, 1H), 2.23 (s, 3H). ^13^C NMR (101 MHz, DMSO) δ 170.4, 169.3, 141.9, 139.4, 139.3, 136.4, 136.2, 132.0, 131.4, 131.1, 129.5, 128.7, 128.6, 128.5, 128.2, 128.1, 128.0, 127.4, 126.8, 115.2, 44.5, 21.0. HRMS (ESI) *m/z* calcd for C_31_H_25_NO_3_ [M+H]: 460.1913, found: 460.1914.

**4b**: Light yellow solid, 364.4 mg, 80% yield (Eluent: dichloromethane/methyl alcohol = 100/1). ^1^H NMR (400 MHz, DMSO) δ 12.90 (s, 1H), 7.83 (dd, *J* = 7.7, 1.5 Hz, 1H), 7.42 (d, *J* = 1.6 Hz, 1H), 7.35–7.27 (m, 5H), 7.30–7.21 (m, 5H), 7.23–7.12 (m, 4H), 7.14–7.07 (m, 2H), 5.61 (d, *J* = 4.3 Hz, 1H), 5.12 (d, *J* = 9.1 Hz, 1H), 4.39 (dd, *J* = 9.2, 4.3 Hz, 1H), 1.19 (s, 9H). ^13^C NMR (101 MHz, DMSO) δ 170.5, 169.3, 149.0, 142.4, 142.0, 139.4, 136.7, 136.5, 132.0, 131.1, 131.1, 128.9, 128.5, 128.3, 128.2, 128.2, 127.9, 127.4, 127.2, 125.3, 115.1, 45.0, 34.6, 31.5. HRMS (ESI) *m/z* calcd for C_34_H_31_NO_3_ [M+H]: 502.2382, found: 502.2381.

**4c**: Light yellow solid, 403.7 mg, 77% yield (Eluent: dichloromethane/methyl alcohol = 100/1). ^1^H NMR (400 MHz, DMSO) δ 13.05 (s, 1H), 7.84 (dd, *J* = 8.0, 1.5 Hz, 1H), 7.47–7.36 (m, 3H), 7.34–7.27 (m, 4H), 7.30–7.19 (m, 6H), 7.22–7.11 (m, 4H), 5.63 (d, *J* = 4.0 Hz, 1H), 5.11 (d, *J* = 9.7 Hz, 1H), 4.44 (dd, *J* = 9.7, 4.0 Hz, 1H). ^13^C NMR (101 MHz, DMSO) δ 170.4, 169.2, 142.4, 141.4, 139.3, 138.8, 136.2, 132.0, 131.6, 131.2, 131.1, 130.9, 128.9, 128.6, 128.4, 128.3, 128.1, 127.4, 127.3, 119.5, 115.6, 45.0. HRMS (ESI) *m/z* calcd for C_30_H_22_BrNO_3_ [M+H]: 524.0861, found: 524.0860.

**4d**: White solid, 374.7 mg, 74% yield (Eluent: dichloromethane/methyl alcohol = 100/1). ^1^H NMR (400 MHz, DMSO) δ 13.10 (s, 1H), 7.84 (d, *J* = 7.7 Hz, 1H), 7.43–7.33 (m, 3H), 7.28 (dd, *J* = 14.6, 8.5 Hz, 8H), 7.18 (t, *J* = 7.1 Hz, 1H), 7.11 (d, *J* = 8.1 Hz, 2H), 7.02 (t, *J* = 8.6 Hz, 2H), 5.61 (d, *J* = 4.2 Hz, 1H), 5.10 (d, *J* = 9.2 Hz, 1H), 4.45–4.37 (m, 1H), 1.20 (s, 9H). ^13^C NMR (101 MHz, DMSO) δ 170.3, 169.3, 163.0, 160.6, 149.2, 142.4, 141.0, 139.4, 136.6, 133.0, 132.9, 132.0, 131.2, 131.1, 130.2, 130.1, 128.9, 128.3, 128.2, 127.4, 127.2, 125.4, 115.5, 115.3, 115.1, 45.0, 34.7, 31.5. ^19^F NMR (376 MHz, DMSO) δ -113.86. HRMS (ESI) *m/z* calcd for C_34_H_30_FNO_3_ [M+H]: 520.2288, found: 520.2286.

**4e**: Light yellow solid, 408.1 mg, 79% yield (Eluent: dichloromethane/methyl alcohol = 100/1). ^1^H NMR (400 MHz, DMSO) δ 13.07 (s, 1H), 7.83 (dd, *J* = 7.8, 1.4 Hz, 1H), 7.46–7.23 (m, 9H), 7.19 (dd, *J* = 10.5, 7.6 Hz, 3H), 7.10 (d, *J* = 8.3 Hz, 2H), 7.00 (d, *J* = 7.9 Hz, 2H), 5.57 (d, *J* = 4.4 Hz, 1H), 5.11 (d, *J* = 8.9 Hz, 1H), 4.35 (dd, *J* = 9.0, 4.4 Hz, 1H), 2.18 (s, 3H), 1.20 (s, 9H). ^13^C NMR (101 MHz, DMSO) δ 170.5, 169.3, 149.0, 141.9, 139.4, 137.5, 136.8, 133.6, 132.0, 131.2, 129.1, 128.9, 128.2, 128.1, 127.8, 127.4, 127.2, 125.4, 114.6, 44.9, 34.7, 31.5, 21.1. HRMS (ESI) *m/z* calcd for C_35_H_33_NO_3_ [M+H]: 516.2539, found: 516.2541.

**4f**: Light yellow solid, 384.3 mg, 78% yield (Eluent: dichloromethane/methyl alcohol = 100/1). ^1^H NMR (400 MHz, DMSO) δ 13.12 (s, 1H), 7.95 (s, 1H), 7.84 (dd, *J* = 12.0, 6.7 Hz, 2H), 7.81–7.75 (m, 1H), 7.68 (d, *J* = 8.6 Hz, 1H), 7.50–7.15 (m, 15H), 7.04 (t, *J* = 7.3 Hz, 1H), 5.76 (d, *J* = 4.2 Hz, 1H), 5.20 (d, *J* = 9.5 Hz, 1H), 4.49 (dd, *J* = 9.6, 4.2 Hz, 1H). ^13^C NMR (101 MHz, DMSO) δ 170.4, 169.3, 142.5, 141.9, 139.4, 139.4, 134.0, 133.0, 132.6, 132.0, 131.2, 131.1, 129.0, 128.8, 128.6, 128.4, 128.3, 127.8, 127.7, 127.4, 127.3, 127.1, 126.9, 126.8, 125.8, 115.7, 45.1. HRMS (ESI) *m/z* calcd for C_34_H_25_NO_3_ [M+H]: 496.1913, found: 496.1916.

## 4. Conclusions

In summary, we developed two efficient transformations of versatile propargyalmines by regulating the reaction conditions. Highly selective cyclization catalyzed via Pd(OAc)_2_ and isomerization promoted by Bu_4_NOAc from propargylamines was successfully implemented. The strategy characterized by readily available starting materials, operational simplicity, mild conditions, broad functional group tolerance, excellent atom economy and high yields is an important advancement in the development of propargylamine-based synthetic methodology. In addition, we believe that this study reveals a new way to prepare nitrogen-containing heterocycles from simple building blocks of amine, aldehyde and alkyne. Further applications of these diverse quinolines and 1-azadienes are underway in our laboratory.

## Data Availability

Not applicable.

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
