# Peer review of "Highly Selective Cyclization and Isomerization of Propargylamines to Access Functionalized Quinolines and 1-Azadienes"

_molecules, 2023, doi:10.3390/molecules28176259_

Round 1

Reviewer 1 Report

In this article, Tang et al. describe a Pd(OAc)2-catalyzed cyclization of propargylamine to synthesize quinolines and a base-promoted isomerization of propargylamide to produce 1-azadienes. While these two types of reactions have been well-documented in the literature, the use of palladium and the milder base TBOAc is a novel approach that has not been reported before. Furthermore, the authors clearly demonstrate the synthetic utility of these two reactions through their application. Taking all these factors into consideration, I recommend accepting this article for publication after minor revisions. Specific suggestions for improvement are provided below.

  1. Title: The current title is too general and could be mistaken for a review article. I suggest that the authors make the title more specific to better reflect the content of the article.

  2. Introduction: I recommend that the authors provide additional information and a figure summarizing previous work on metal-catalyzed cyclization of propargylamine to synthesize quinolines and base-promoted isomerization of propargylamide to produce 1-azadienes. This will help to highlight the differences between previous work and the work presented in this article.

Minor errors:

  1. Line 132: The description of compound 2o as an “aliphatic aldehyde” is not accurate. I suggest using “aliphatic group, i.e. cyclohexyl” instead. Similarly, for “heteroarene,” it would be more specific to use “heteroarene, i.e. 2-thiophenyl.”

  2. Scheme 2a, compound 2m: The yield is given in parentheses, which could be interpreted as a gram-scale yield. I suggest adding a footnote for clarification.

  3. Scheme 3, Intermediate B1: It would be clearer to add a dot to represent the carbon atom in the middle of the allenic intermediate.

Author Response

Comments 1: Title: The current title is too general and could be mistaken for a review article. I suggest that the authors make the title more specific to better reflect the content of the article.

Response: Thank you for the valuable comments. The title has been changed to “Highly selective cyclization and isomerization of propargylamines to access functionalized quinolines and 1-azadienes”.

Comments 2: Introduction: I recommend that the authors provide additional information and a figure summarizing previous work on metal-catalyzed cyclization of propargylamine to synthesize quinolines and base-promoted isomerization of propargylamide to produce 1-azadienes. This will help to highlight the differences between previous work and the work presented in this article.

Response: Thank you for the valuable comments. Different synthesis strategies of quinolines and 1-azadienes from propargylamines have been summarized in the revised manuscript (Scheme 1).

Minor errors:

  1. Line 132: The description of compound 2o as an “aliphatic aldehyde” is not accurate. I suggest using “aliphatic group, i.e. cyclohexyl” instead. Similarly, for “heteroarene,” it would be more specific to use “heteroarene, i.e. 2-thiophenyl.”
  2. Scheme 2a, compound 2m: The yield is given in parentheses, which could be interpreted as a gram-scale yield. I suggest adding a footnote for clarification.
  3. Scheme 3, Intermediate B1: It would be clearer to add a dot to represent the carbon atom in the middle of the allenic intermediate.

Response: Thank you for your constructive suggestions. The inappropriate contents have been corrected and improved accordingly in the revised manuscript, which are highlighted in yellow.

Author Response

Comments: This manuscript is the report on two transformations of A3 coupling products into quinoline heterocycles and 1-azadienes using palladium(II) catalyst and acetate ion, respectively. Because there are a lot of papers on the former transformations using other metal catalysts including Ag, Cu, Fe, and Hg, I don't understand whether the reaction condition the authors have developed in this article has benefit or not. The most problem is that they have not cited these papers (Org. Chem. Front. 2018, 5, 434–441; Chem. Lett. 2007, 36, 1422–1423; Org. Biomol. Chem. 2014, 12, 255–260; Adv. Synth. Cat. 2014, 356, 692–696; J. Fluorine Chem. 2012, 135, 195–199; J. Fluorine Chem. 2012, 135, 139–145; Tetrahedron 2010, 66, 1177–1187). Furthermore, I'm also wondering whether the reaction condition for the later transformation is superior to the previously reported one (ref 35). The later transformation is quite different from the former one and should not be combined in a single paper. At this moment, the paper should be rejected as Article in Molecules.

Response: Thank you for your comments. While these two types of reactions have been widely reported, the use of palladium and the milder base TBOAc is a novel approach, which benefits from easily accessible precursors, mild conditions, excellent selectivity and flexible late-stage functionalization, illustrating the high efficiency and synthetic utility of this highly selective transformation of propargylamines.

In the previous manuscript, we just cited some representative literatures for quinoline heterocycles formation from propargylamines. We are sorry for our carelessness and other papers recommended by reviewer have been supplemented to the revised manuscript.

The previous work (ref 35) exhibits relatively complex reaction system using superbase (KOBut)/DMSO/THF, while the milder TBOAc-promoted isomerization is easy for further functionalization. In this manuscript, 1-azadienes in situ synthesized from propargylamines could be directly used to prepare a series of medicinally important dihydropyridin-2(1H)-ones (4a-4f) via a [4+2]-formal cycloaddition reaction with homophthalic anhydride through a one-pot model, illustrating the validity and practicability of the propargylamine-based highly selective reactions.

Reviewer 3 Report

Please consider the attached file.

Author Response

Comments 1: Since the authors have employed diverse propargylamines instead of only propargylamine, please consider modifying the article title accordingly. As a suggestion, it would be great to also emphasize what was achieved in the title (E.g Highly selective cyclization and isomerization of propargylamines to access dihydropyridin-2(1H)-ones and 1-azadienes).

Response: Thank you for your valuable comments. The title has been changed to “Highly selective cyclization and isomerization of propargylamines to access functionalized quinolines and 1-azadienes”.

Comments 2: Supporting Information, NMR data. Please report the chemical shifts with one decimal precision for 13 C NMR data (SI). Additionally, please include NC within parenthesis when an observed chemical shift is related to N carbons being N>1 (Please do so for all related instances. N stands for number).

Response: Thank you for the valuable comments. We have revised them accordingly.

Comments 3: What are the advantages of the devised synthesis of 1-azadienes with TBOAc over KOBu t (DOI: 10.1021/acs.joc.9b03192)? How does the Pd-catalyzed synthesis of quinolines from propargylamines compare to the use of FeCl3 in a one-pot manner (10.1002/chem.200900875)?

Response: Thank you for your valuable comments. The previous work (10.1021/acs.joc.9b03192) by Trofimov exhibits relatively complex reaction system using superbase (KOBut)/DMSO/THF, while the milder TBOAc-promoted isomerization is easy for further functionalization, even without separation of TBOAc. For example, in this manuscript, 1-azadienes in situ synthesized from propargylamines could be directly used to prepare a series of medicinally important dihydropyridin-2(1H)-ones through a one-pot model, illustrating the validity and practicability of the TBOAc-promoted propargylamine transformation. In addition, the reaction using TBOAc exhibited higher yield up to 93% than that of KOBut (up to 72%).

    The FeCl3-catalyzed three-component tandem reaction of aldehydes (except for aliphatic aldehyde), alkynes, and amines (10.1002/chem.200900875) is a facile and economic method for the construction of quinolines, while Pd-catalyzed synthesis of quinolines from propargylamines also affords an alternative strategy with good functional group tolerance, mild condition, excellent atom economy, et al. Both two reactions could promote the development of propargylamine-based synthetic methodology.

Comments 4: Page 4. Lines 131-135. The mentioned codes do not refer to the used starting materials but to the final products, quinolines (E.g. 2o directly refers to the quinoline bearing a cyclohexane.) Please make some adjustments to this excerpt while mentioning the starting materials.

Response: We are sorry for our carelessness and we have revised them accordingly.

Comments 5: How challenging was the purification of the prepared 1-azadienes? According to the provided related NMR spectra, there are some impurities. Did the authors measure the ratio between the possible isomers? Was some selectivity achieved?

Response: Thank you for your valuable comments. Due to the effective conversion and high yield, the purification of the target 1-azadienes is relatively easy, which could be directly obtained through simple flash column chromatography. Some impurities may be attributed to the slow decomposition of 1-azadienes exposed to the solution and air, for some special structures.

From the NMR spectra, for example, the azadiene 3a in CDCl3 presents as a mixture of 1E,2E- and 1Z,2E-isomers in a 3:1 ratio, which are in accord with the results from Boris A. Trofimov (J. Org. Chem. 2020, 85, 3417−3425).

Comments 6: Page 5, lines 182-184. The authors have emphasized that dihydropyridin-2(1H)-ones find great application in medicinal chemistry. However, no references were provided to sustain this affirmation.

Response: Thank you for your comments. The corresponding references (refs 47, 48) have been supplemented to the revised manuscript.

Round 2

Author Response

Reviewer 2: This manuscript is the report on two transformations of A3 coupling products into quinoline heterocycles and 1-azadienes using palladium(II) catalyst and acetate ion, respectively. Now they have cited papers on the former transformations using other metal catalysts, but there are still two things I concern about. I don't feel that "an amine group in β-position to an alkyne moiety" (Text in Line 29th, Page 1) is appropriate to represent propargylamine. Since TBOAc is not a familiar abbreviation, Bu4NOAc would be better (Text in Line 78, 82, 138, 223, 231, 261th, Table 1, entry 28 and footnote). After minor revision, the paper can be accepted as Article in Molecules.

Response: Thank you for your valuable comments. The inappropriate contents of “an amine group in β-position to an alkyne moiety” have been corrected and changed to “consist of amine groups and alkyne moieties on the same backbone” in the revised manuscript. In addition, TBOAc has been replaced by Bu4NOAc accordingly.